# Effects of Biochar and Manure Co-Application on Aggregate Stability and Pore Size Distribution of Vertisols

**DOI:** 10.3390/ijerph191811335

**Published:** 2022-09-09

**Authors:** Taiyi Cai, Zhigang Wang, Chengshi Guo, Huijuan Huang, Huabin Chai, Congzhi Zhang

**Affiliations:** 1School of Surveying and Land Information Engineering, Henan Polytechnic University, Jiaozuo 454000, China; 2State Experimental Station of Agro-Ecosystem in Fengqiu, State Key Laboratory of Soil and Sustainable Agriculture, Institute of Soil Science, Chinese Academy of Sciences, Nanjing 210008, China; 3Farmland Irrigation Research Institute, Chinese Academy of Agricultural Sciences, Xinxiang 453002, China

**Keywords:** soil health, biochar, manure, soil structure, vertisol

## Abstract

Background: The combination of biochar and organic manure has substantial local impacts on soil properties, greenhouse gas emissions, and crop yield. However, the research on soil health or quality is still in its early stages. Four pot experiments were carried out: C (30 g biochar (kg soil)^−1^), M (10 g manure (kg soil)^−1^), CM (15 g biochar (kg soil)^−1^ + 5 g manure (kg soil)^−1^), and the control (without any amendments). Results: When compared to C and M treatments, the MWD of CM was reduced by 5.5% and increased by 4.9%, respectively, and the micropore volume (5–30 m) was increased by 17.6% and 89.6%. The structural equation model shows that soil structural parameters and physical properties regulate the distribution of micropores (5–30 μm) in amended soil. Conclusion: Our studies discovered that biochar mixed with poultry manure had antagonistic and synergistic effects on soil aggregate stability and micropore volume in vertisol, respectively, and thus enhanced crop yield by 71.1%, which might be used as a technological model for farmers in China’s Huang-Huai-Hai region to improve low- and medium-yielding soil and maintain soil health.

## 1. Introduction

Soil health is defined as soil’s ongoing ability to support vital ecosystem functions for plants, animals, and humans [1]. Soil health or quality has become a hot topic in research on sustainable agricultural development today, and it can be evaluated quantitatively and qualitatively through soil health indices (soil processes and traits) [2]. Soil aggregate stability and soil pore characteristics, for example, are important indicators for characterizing soil health and the sustainability of various agricultural management practices [3,4].

Aggregate stability refers to the ability of soil aggregates to withstand damage when subjected to external forces (such as rain) [5]. It describes the physical ability of soils to maintain their aggregation and structure in wet conditions (such as irrigation after severe rainfall or protracted drought). Aggregate stability is regarded as a reliable indicator of soil’s biological and physical health. According to research, poorly structured soils, such as vertisol, frequently develop surface crusts or compaction, which can impair water transport and gas exchange and increase the risk of flooding and drought. Internal and extrinsic factors both influence soil aggregate stability [6]. Meteorological conditions (e.g., average annual precipitation and temperature) and agricultural management practices (e.g., fertilization and tillage) are examples of external influences, while intrinsic factors include soil clay content, organic matter content, and soil minerals [7].

Soil pores are the spaces between soil particles or aggregates. Pore properties are the “leading indicator” for determining many soil processes and functions, including water storage and transport, microbial activity, and soil mechanical resistance to root penetration [8]. Soil pore size distribution (PSD) has been shown in studies to be an important indicator for a thorough understanding of soil aggregate stability, water transport, and carbon sequestration [9]. Many studies have shown that fertilization, tillage, and land use can all have a significant impact on the total porosity, size distribution, and function of soil pores [4,10,11], thereby affecting soil quality. This implies that soil pore structure is highly susceptible to soil management practices and environmental changes. As a result, studying soil pore characteristics is vital for assessing soil health.

Organic manure is an efficient way to improve soil health and plant growth [12]. In comparison to inorganic fertilizers, it primarily promotes plant growth in the soil by slowly decomposing and releasing various nutrients. Many types of organic manure on the market are derived from biochar, animal manure, and wood waste compost, and their effects on promoting healthy soils have been thoroughly researched [13,14].

Biochar is a byproduct of high-temperature pyrolysis of natural organic materials that are rich in carbon and porous materials [15]. Currently, the primary application of biochar in agriculture is as a soil amendment [16]. In-depth research has demonstrated that biochar may improve soil fertility, detoxify the soil, promote soil microbial diversity, and improve plant health, and it is projected to become one of the main efforts to preserve soil health and sustainable agricultural development. However, biochar prices are high and cannot be offset by prospective economic gains based on greater average yields and current CO_2_ pricing [17]. As a result, biochar has yet to be used on a broad basis in agricultural operations [18,19].

Livestock and poultry waste are the primary sources of animal manure. Poultry manure, in particular, is widely used as a soil improvement material due to its obvious advantages of high nitrogen content, low price, and high yield [20], which can increase the source of soil nutrients and organic matter and can improve the soil microbial population, affecting soil structure and soil physical properties, as well as chemical and biological changes in other parameters [21]. According to a literature study, the amount of manure generated by large-scale chicken dung in China is 3.83 × 10^9^ t, with Henan Province producing the most, followed by Shandong, Hebei, Sichuan, and Hunan provinces [22]. This suggests that poultry manure has a high potential for organic manure generation in soil.

Vertisol is soil with a similar mineral composition and characteristics to its parent material [23]. It is commonly available in Australia, China, India, the United States, and other countries [24]. Deep cracks and sticking behavior are easily developed as a result of the continual expansion and shrinkage of clay-rich minerals, posing a possible threat to agricultural production. However, due to its potential high natural fertility, it has piqued the interest of many scholars both domestically and internationally. Vertisols cover approximately 4 × 10^6^ ha in China, with the majority of them located in semi-arid northern China and belonging to low- and medium-yielding soils [4]. To enhance its poor physical structure, predecessors attempted to implement various technologies (organic fertilizer, fly ash, biochar, and amendment, among others) and achieved considerable progress [3,4].

Previous research has mostly focused on the single effect of organic manure or biochar on the structural improvement of vertisols, focusing little on the combined effect of the two. Several studies combining the two have shown promising results in terms of soil C and N cycling processes, soil biological indicators (soil microbial biomass, enzyme activity, and soil microbial diversity), and crop yield enhancement [14,25,26], but there has been a severe lack of studies on the effects of the two on soil structure. Due to the uniqueness and typicality of the vertisol soil structure, improving its poor physical structure has always been the primary goal of vertisol research.

As a result, it is critical to understand the impact of the two on the physical structure of the modified soil. Through pot experiments, the goal of this study is to investigate the effects of combining biochar and organic fertilizer on soil aggregate stability, soil pore structure and their combined response (crop yield and economic profit). When compared to biochar or organic manure treatments alone, we hypothesized that co-applying biochar and organic manure improves soil’s aggregate stability and optimizes soil pore size distribution in vertisols, which in turn will increase crop yield and economic efficiency, thus maintaining soil health.

## 2. Materials and Methods

### 2.1. Potted Experiment Method

The experimental soil was originally taken from a typical vertisol of a 0–30 cm soil layer in Dancheng County, Zhoukou City, Henan Province, China (33°38 N, 115°23 E, elevation of 23 m). Large aggregates were crushed, and debris, such as rocks larger than 3 cm in diameter, was removed. The soil is classified as a typical vertisol by the Chinese soil classification method [27]. Before the test, the selected soil physicochemical characteristics were determined using the Zhang and Gong [28] test method, and the results are shown in Table 1.

The test pots were 52 cm tall, with an upper inner diameter of 44 cm and a lower diameter of 30 cm, and a water outlet hole at the bottom. Each pot was filled with 70 kg of drying soil with a filling capacity of 1.3 g cm^−3^.

The experiment started in 2012. Given the characteristics of poor soil physical properties (soil viscosity, high bulk density, and poor ventilation and water permeability) of sand ginger black soil, a total of four treatments were set up in a completely randomized block design and repeated four times. The four treatments were as follows: C (30 g biochar (kg soil)^−1^), M (10 g manure (kg soil)^−1^, CM (15 g biochar (kg soil)^−1^ + 5 g manure (kg soil)^−1^), and the control (without any amendments). These amounts refer to the ranges generally applicable in the region and other countries [4,14]. After the crops were harvested, the soil for each treatment was removed from the pot and sieved, as biochar is inert and basically does not decompose, and the biochar treatment was no longer required. Each pot of organic manure treatment was added again according to the soil ratio of 1.04 g/kg, and it was used for potting experiments in the summer. The applied crop rotation method was winter wheat–summer corn.

Wheat (*Triticum aestivum* L.) was planted on 9 October and harvested on 1 June; maize *(Zea mays* L.) was planted on June 10 and harvested on 1 October.

Biochar was created in a factory-scale reactor by pyrolysis at 500–550 °C for 20 h, then sieved to 2 mm and analyzed (Table 2) according to the methodology recommended by Xie et al. [29]. Poultry manure was composted at 30–70 °C for 30 days and kept above 55 °C for 7 days before its physicochemical properties were determined using the method recommended by Xie et al. [29].

Wheat seeds were sown in each pot in four rows. Maize was sown in pots with 10 seeds per pot, and four plants were kept until harvest. To calculate yield, the aboveground parts of the plants (straw and grain) were harvested separately and dried at 70 °C for 72 h.

### 2.2. Laboratory Methods

#### 2.2.1. Soil Samples

In October 2019, after the end of the experiment (after maize harvesting), 15 cm soil samples were collected from each pot, air-dried, and sieved (<2 mm), and physicochemical properties were analyzed according to traditional methods [28].

#### 2.2.2. Determination of Physical Properties

(1) Coefficient of linear expansion (COLE)

The COLE of soil was determined on ground remolded soils according to Schafer and Singer [30]. The COLE was calculated using the formula below:COLE = (Lm − Ld)/Ld 
where Lm and Ld are the length of moist and dry soils, respectively.

(2) Distribution and stability of soil aggregate

Soil agglomeration distribution was performed using the wet sieve method. For the main steps, 50 g of soil samples was weighed and placed on the top of a set of sieves with diameters of 2, 1, 0.5, and 0.25 mm, and all the sieves were placed in an agglomerate shaker model (DM200-II) and shaken up and down at a frequency of 30 times per minute for 30 min. Finally, the remaining agglomerates of each sieve were dried in a drying oven at 105 °C for 24 h and weighed, and the percentage of soil water stability mass for each particle size was calculated. The calculation formula of soil aggregate stability (average weight diameter) is as follows [31]:$$MWD=\sum_{i}^{n}X_{i}W_{i}$$

In the formula, *MWD* is the average weight diameter, *X_i_* is the grain size average diameter (mm); *W_i_* size *X_i_* is the body weight percentage.

#### 2.2.3. Pore Size Distribution (PSD)

Using a mercury porosimeter (MIP) model, PSD was calculated (Autopore IV9500, Micromeritics Inc., Norcross, GA, USA). The mercury porosimetry method is based on the hypothesis that the soil pores are small, irregularly shaped, and cylindrical. Each pore can extend to the sample’s outer surface and come into direct contact with the mercury (theta is 140°). The Washburn formula is as follows:$$r=\frac{2\gamma \cos\theta }{p}$$

The pores were assumed to be cylindrical, where *r* is the pore radius (μm), p is the pressure (kPa), γ is the Hg surface tension (0.47 N m^−1^), and θ is the mercury–soil contact angle (140°). Under certain pressure, the mercury will infiltrate into pores of the corresponding size, and the amount of indented mercury represents the volume inside the pores; if the pressure is gradually increased, the amount of indented mercury can be calculated, and the volume distribution of soil pores can be measured [32]. To gain an in-depth understanding of soil pore size distribution, soil pores are divided into five levels: macropores (>60 μm), mesopores (60–30 μm), micropores (30–5 μm), ultra-micropores (5–0.1 μm), and crypto pores (0.1–0.01 μm and <0.01 μm) [33].

#### 2.2.4. Determination of Chemical Properties

The pH was determined using a standard method, i.e., a soil-to-water ratio of 1:2.5, and the mixture was shaken uniformly after 60 min of settling; subsequently, the pH of the suspension was measured using a pH meter [28].

Total organic carbon was determined using the C oxidation method with potassium dichromate, followed by the titration of the remaining Cr_2_O_7_^2−^ with ammonium iron (II) sulfate [34].

### 2.3. Statistical Analysis

The multivariate statistical analysis of soil pore types and soil physical and chemical properties was conducted using the SPSS 21.0 Statistical Package (Tsinghua University Press, Beijing, China). Structural equation modeling (SEM) is a multivariate statistical method that can perform hypothesis testing on complex path relationships between indicators [35]. Based on the relative contribution rates of biochar and organic manure to soil micropore volume and their interaction, we used AMOS 21.0 software (IBM, Armonk, NY, USA) for structural equation modeling and quantitative analysis. Drawing using Sigmaplot12.5.

## 3. Results

### 3.1. Selected Physical and Chemical Properties of Vertisol Soil

The physical and chemical properties of vertisol soil selection are shown in Table 3. The soil has typical alkaline vertisol characteristics (pH > 7.98). The pH values of C, M, and CM treatments were not significantly different, and the pH values of the three treatments were all lower than those of the control. Soil organic carbon (SOC) ranged from 22.47 to 27.33 g kg^−1^. On average, the soil organic carbon (SOC) in C (27.33 g kg^−1^), M (26.87 g kg^−1^), and CM (26.61 g kg^−1^) were significantly higher than in the control (22.47 g kg^−1^), respectively. The COLE ranged from 0.09 to 0.12, with an average of 0.105. The COLE values of the C and CM treatments were significantly lower than that of the control and M treatments; however, there was no significant difference between the C and CM treatments.

### 3.2. Soil Aggregates

Figure 1 shows the improved vertisol aggregate size distribution with biochar and poultry manure. Biochar and poultry manure changed the soil aggregate properties of vertisol. Compared with the control, the percentage of water-stable macroaggregates of 2–0.25 mm was significantly increased for the C, M, and CM treatments, while the percentage of microaggregates < 0.053 mm was significantly decreased; however, the percentage of soil aggregates > 2 and 0.25–0.053 mm did not differ among the three treatments. Meanwhile, the CM treatment (biochar and manure co-application) had the highest percentage of 2–0.25 mm aggregates (26.3%), while the percentage of <0.053 mm aggregates was the lowest (61.9%), which was increased by 28.7% compared with the control, which decreased by 9.4%. This indicates that the interaction of biochar and poultry manure upon aggregate formation was significant (*p* < 0.05).

The mean weight diameter (MWD) plays an important role in assessing the stability of soil aggregates, with large MWD values indicating better water stability of aggregates [36]. Compared with the control, the MWD of the C, M, and CM treatments was significantly improved, and the CM treatment was the most effective (Figure 1). The MWD of CM was 4.9% higher than that of the M treatment, but 5.5% lower than that of the C treatment, which indicated that the combination of C and M would have obvious antagonistic effects, that is, reducing the positive effect of biochar treatment and enhancing the negative effect of poultry manure treatment.

### 3.3. Pore Characteristics and PSD of Soil

#### 3.3.1. Pore Size Distribution

The cumulative pore volume and differential pore size distribution (PSD) curves of the vertisol soil show that (Figure 2) the PSD curve of vertisol has a unimodal distribution overall. The peak value (0.12–0.30 μm) was shifted to the right, and when the equivalent pore size was <0.01 μm, there was no significant difference between the two pore size distribution curves. The PSD changes in the C and CM treatments were similar. Compared with the control and M treatments, their PSD distribution curves shifted to the right as a whole, with peaks at 12–16 μm and 50–60 μm, respectively. The CM peak was significantly higher than that of the C treatment, which indicated that the CM treatment had a significant positive effect in increasing the pore volume with pore sizes of 5–30 μm; however, when the equivalent aperture was <5 μm, there was no significant difference in PSD distribution between the two treatments.

#### 3.3.2. Pore Volume Distribution

Pore size distribution (PSD) data can be used to characterize soil evolution and describe agricultural management effects [37,38]. The soil pore volume distribution and total pore volume determined based on the MIP method are shown in Figure 3A. The total pore volumes of soils treated with C and CM were 0.2859 and 0.2886 cm^3^ g^−1^, respectively, which were significantly higher than (*p* < 0.05) that of the control soils (0.1261 cm^3^ g^−1^), while the total pore volumes of soils treated with M (0.1014 cm^3^ g^−1^) were significantly lower than the control. Compared with the control, C, and M treatments, the total pore volume of the CM treatment was increased by 128.9%, 0.9%, and 184.6%, respectively, which indicated that the CM treatment reflected the synergistic effect of the mixing of biochar and poultry manure.

Figure 3B showed that the C, M, and CM treatments significantly reduced the volume percentage of macropores by 100–60 μm and significantly increased the volume percentage of micropores by 5–30 μm compared to the control, which further indicated the biochar and poultry manure in the pores. The improvement of the structure had a significant synergistic effect; however, the difference was that there was a positive synergistic effect on the 5–30 μm pore volume percentage, but a negative synergistic effect on the 100–60 μm pore volume percentage. Changes in the volume percentage of pores at other grades (60–30, 5–0.1, and <0.1 μm) were irregular.

### 3.4. Correlation between Soil Micropore Characteristics and Soil Physicochemical Properties

Pearson correlation coefficients were calculated to describe the correlation of micropore volume changes with soil properties and composition (Table 4). There was a significant positive correlation between the micropore volume (P5–30) and TPV, porosity, fractal dimension, and A2–0.25 (*p* < 0.05), but a significant negative correlation with P0.1–5, P0.01–0.1, *p* < 0.01, pH, COLE, and A < 0.053 (*p* < 0.05).

### 3.5. Structural Equation Modeling

Due to the strong correlation between the micropore volume (P5–30) and physicochemical parameters, we introduced structural equation modeling to deeply explore the mechanism of biochar and livestock manure on the change in micropore volume (Figure 4). According to the three indicators of the model fitting effect (*χ*^2^ = 6.437, *df* = 6, and *p* = 0.368), the SEM model constructed in this study can better fit the internal relationship between the relevant indicators, assuming that causal variables explain 73.5% of the reason for the change in pore volume (5–30 μm).

The results of the optimal model fitting (Chi-square (*χ*^2^) = 23.46; *df* = 12, *p* = 0.0481; comparative fit index (CFI) = 0.918; root square mean error of approximation (RMSEA) = 0.221). Square boxes denote variables included in the models. The solid arrows (→) mean a single effect in the direction of the arrow, the lines (-) mean a cycle effect, and the thickness represents the magnitude of the path coefficients. Dashed arrows represent the directions, and the effects were non-significant (*p* > 0.05). (Porosity and total MIP porosity for the pore diameter range of 0.003–360 μm; FD, fractal dimension; COLE, linear expansion coefficient; Percentage of A2–0.25 and 2–0.25mm aggregates; Percentage of A < 0.053 and <0.053 mm aggregates; and P5–30 and 5–30 μm, soil pore volume.)

The structural equation model demonstrated that porosity, fractal dimension, and 2–0.25 mm aggregates had direct positive effects on the 5–30 μm pore volume, and porosity and fractal dimension had indirect positive effects on the 5–30 μm pore volume through 2–0.25 mm soil aggregates, respectively. COLE and <0.053 mm soil aggregates had a direct negative effect on the 5–30 μm pore volume, and COLE produced an indirect negative effect on the 5–30 μm pore volume through <0.053 mm soil aggregates.

### 3.6. Grain Yield and Economic Profit

Table 5 shows that the grain yields of C, M, and CM (0.37, 0.29, and 0.33 kg (Per Plant)^−1^) were increased by 92.8%, 49.9%, and 71.1%, respectively, compared with the control (0.19 kg (Per Plant)^−1^). Among them, the yield of CM was between C and M. This suggested that the combination of biochar and organic manure had an antagonistic effect on maize grain yield. In terms of economic Profit, the M treatment had the highest economic Profit, followed by the control treatment, and the C and CM treatments were the lowest.

## 4. Discussion

### 4.1. Amelioration of the Aggregate Stability

Soil aggregate stability is an indicator for assessing the effects of soil type and field management on soil quality [6]. The biochar-amended soil significantly increased the formation of 2–0.025 mm aggregates while decreasing the percentage of 0.053 mm microaggregates (Figure 1), indicating that the macroaggregates were formed through the combination of many microaggregates [31]. The biochar-enhanced carbon may act as a glue, concentrating microaggregates into macroaggregates. Our findings are consistent with those of Lu et al. [39], who discovered that rice husk biochar-amended soils have a significantly higher MWD than the control soils [36]. The improvement in aggregate stability was thought to be the result of two mechanisms. The first is that biochar improves soil properties through the physical meshing of carbon polymers or particles, increasing the internal cohesion of mineral particles to increase aggregate resistance to clay swelling [40], and the second is that biochar treatment promotes soil hydrophobicity properties, thereby reducing the degree of clay swelling and aggregate breakage [3]. The stability of aggregates in this study could be due to the interaction of these two mechanisms.

Several studies have suggested that adding poultry manure to soil improves MWD [41,42]. The use of poultry manure (2% by weight) increased wet aggregate stability (Figure 1). The MWD value of the poultry manure treatment was slightly higher than that of the control, but the difference was not significant, confirming a previously established effect of poultry manure on soil aggregation stability. This finding is consistent with that of Peng et al. [43], who concluded that adding swine manure increased dry aggregate stability and Peanut biomass in a Ultisol. Figure 1 also shows that biochar co-application with manure treatment significantly increased the percentage of 2–0.25 mm aggregate and MWD when compared to the C and M treatments, indicating a positive synergy effect between C and M. These findings are consistent with those of Sánchez et al. [41], who discovered that the addition of 3% biochar promoted the rapid degradation of organic matter, reduced the formation of large clumps, and accelerated stabilization and detoxification. One possible explanation for this is that biochar can help increase the carbon source for microorganisms or promote the rapid degradation of poultry manure [42].

### 4.2. Amelioration in the Soil Pore System

Soil pore characteristics (such as porosity, size distribution, and Shape of soil pore space) are important indicators of soil quality and are highly sensitive to soil management practices, such as biochar and manure application in the field [8,39]. Clay PSDs usually have more micropores (<0.01 μm). The 100–0.3 μm pore volume was significantly higher in the C and CM treatments than in the control and manure treatments (Figure 3). This variation could be attributed to the inherent properties of biochar (e.g., wider pore size distribution) [44]. Other authors have reported that organic waste-based biochar contains many macropores larger than 10 μm in diameter, which induce soil particle cohesion and thus improve the pore structure [45].

The use of poultry manure has been shown to improve soil structural stability [43,46]. In the current study, CM reduced soil porosity and was consistent with the changing trend of the control treatment; however, the pore volume fraction at a 0.5 m equivalent pore diameter was higher than that of other treatments (Figure 3B).

This finding contradicts previous research, which has found that organic manure treatment increases Porosity more than the control [14,47]. This result could be explained in two ways. On the one hand, the decrease in total pore porosity in soil may be due to an increase in ultramicropores (0.1–5 m) and crytopores (0.1–0.007 m), which have become the dominant component of the pore system. On the other hand, it is likely that changes in the composition and arrangement of soil particles (or aggregates) caused a decrease in porosity [46]. However, it is unknown how the composition and arrangement of soil particles (or aggregates) affect the decrease in total pore porosity; therefore, more research should be conducted.

Figure 3 and Figure 4 show that the co-application of biochar with manure had a significant positive synergistic effect on microspore volumes of a 5–30 μm equivalent pore size. The synergistic effect was comparable to adding cow manure (2% by weight) and biochar (10% by weight) to calcareous soils, and it demonstrated a positive priming effect [48]. However, the findings of this study do not support those of Binh Thanh et al., [49] that the combination of biochar and cow manure had no positive synergistic effect. Although the underlying reasons for the opposite conclusion are yet to be determined, the dilution effect caused by the mixing of livestock manure and clay minerals could be one of them [46]. On the one hand, biochar’s ion adsorption creates a favorable solution environment. Meanwhile, biochar accelerates the mineralization of livestock manure [42], Providing more ion adsorption for biochar, but biochar pores typically exceed 10 mm [45]. As biochar can only selectively adsorb parts of ions, it is the collaboration of biochar and manure that produces these synergistic effects. However, currently, this effect mechanism is only speculative, and more research is needed to confirm it.

### 4.3. MicroPore Change Based on Structural Equation Model

Interaction effects between different factors are frequently masked by simple bivariate correlations, and SEM can answer questions involving multiple regression analyses of factors [35]. We conducted a structural equation simulation analysis of soil pore structure parameters and soil micropore volume to reveal the pathway of biochar and organic manure affecting soil micropores (5–30 μm) via changes in soil aggregates and pore structure parameters. In this study, we hypothesized that soil micropores (5–30 μm) were affected by three pathways (Figure 4): the soil pore structure (represented by porosity and fractal dimension); soil aggregates (represented by the number of aggregates between 2–0.25 mm and 0.053 mm); and soil physical properties (represented by COLE), and that any difference in these parameters could explain soil micropore (5–30 μm) volume variation.

Soil porosity and fractal dimension contributed the most to the soil micropore (5–30 μm) volume in the first approach, with contributions of 0.986 and 0.965, respectively, and in the second pathway, the contribution rates of 2–0.25 mm and 0.053 mm aggregates were 0.592 and 0.641, respectively. COLE’s contribution to the soil micropore (5–30 μm) volume was −0.688 in the third approach. It is worth noting that biochar and organic manure had a direct and indirect effect on the volume change in soil micropores (5–30 μm) via porosity, fractal dimension, and COLE, which was consistent with the findings of Guo et al. [50].

According to Table 1, biochar treatment enhanced porosity and fractal dimension while decreasing COLE. The use of organic manure reduced porosity while increasing the fractal dimension and COLE. CM treatment, on the other hand, enhanced porosity and fractal dimension while decreasing COLE. When these findings are combined with the structural equation model, it is obvious that the volume change in soil micropores (5–30 μm) is the result of the synergistic and antagonistic effects produced by the combination of biochar and organic fertilizer. However, given Present technological Capabilities, it is still impossible to quantify the synergistic and antagonistic effects. To this aim, the best microscopic technology must be developed, as well as advanced spectroscopy (NMR) and isotope labeling (^13^C and ^15^N) technologies.

### 4.4. Economic Profit

The combination of biochar and organic manure needs to be verified by comprehensive experiments with cost–benefit methods [18,20]. The most significant improvement effect of biochar and organic manure was an increase in crop yield (Table 5). As this was a pot experiment, only the corn yield was used as an example to facilitate the comparison with the field experiment. The field yield conversion method was as follows: convert the corn yield per plant in the pot experiment into the field yield based on 3500 plants (667 m^2^)^−1^, which is approximately 10,191 kg ha^−1^. The use of biochar in combination with livestock manure increased production by 71.1%. When combined with the current situation of low local farmer income, high biochar costs, and abundant Poultry manure, this combination is recommended as a feasible alternative to increase maize yield in this region while considering appropriate economic benefits.

In general, vertisol improvement in China’s Huang-Huai-Hai plain requires approximately 5 tons ha^−1^ of organic manure, with a total value of approximately CNY 500 [51], and with a local maize price of CNY 1.5/kg^−1^ [52], the application of organic manure would yield CNY 15316. Given that the cost of biochar varies and that the price in the region is approximately CNY 1800/t^−1^ [17], the benefit of 20 t biochar was approximately CNY 6379, which was lower than the benefit of organic manure alone and the control. However, as biochar is a major source of inert carbon with a long average retention time (decades to centuries), it may have a long-term impact on soil quality [20].

## 5. Conclusions

The effects of biochar, organic manure, and a combination of the two on soil properties and crop growth were significant. The combination (15 g biochar (kg soil)^−1^ + 5 g manure (kg soil)^−1^) demonstrated antagonistic effects on aggregate stability and crop yield, as well as synergistic effects on soil macropore and Micropore volume distribution, with the former being negative and the latter being positive.

Combining biochar and organic manure application can be a viable management practice for smallholder farmers in China’s Huang-Huai-Hai Plain region, as it not only improves aggregate stability and the distribution of soil micropores (5–30 μm) corresponding to plant available water, but also reduces COLE and soil improvement costs, increases crop yields, and thus maintains soil health when compared to applying biochar or organic manure alone. Further research will be required to accurately match the ratio of biochar to organic manure to apply this improved technology more widely.

## Figures and Tables

**Figure 1 ijerph-19-11335-f001:**
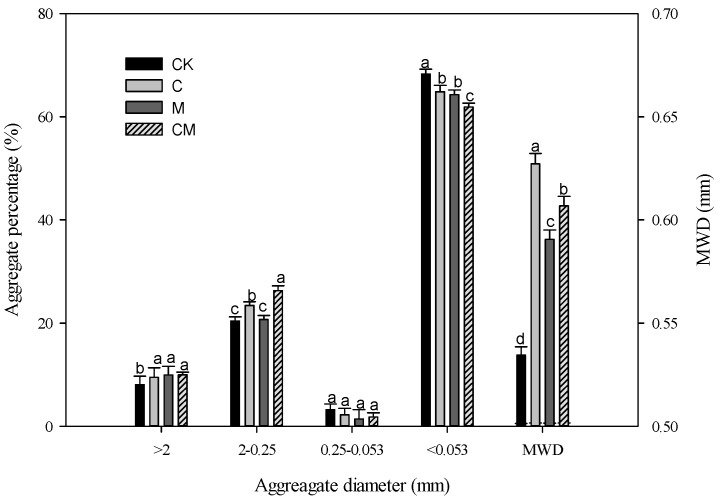
Effects of different treatments on the vertisol aggregate size distribution using wet sieving method and MWD, where error bars represent standard deviation and lowercase letters indicate significant differences between treatments (*p* < 0.05).

**Figure 2 ijerph-19-11335-f002:**
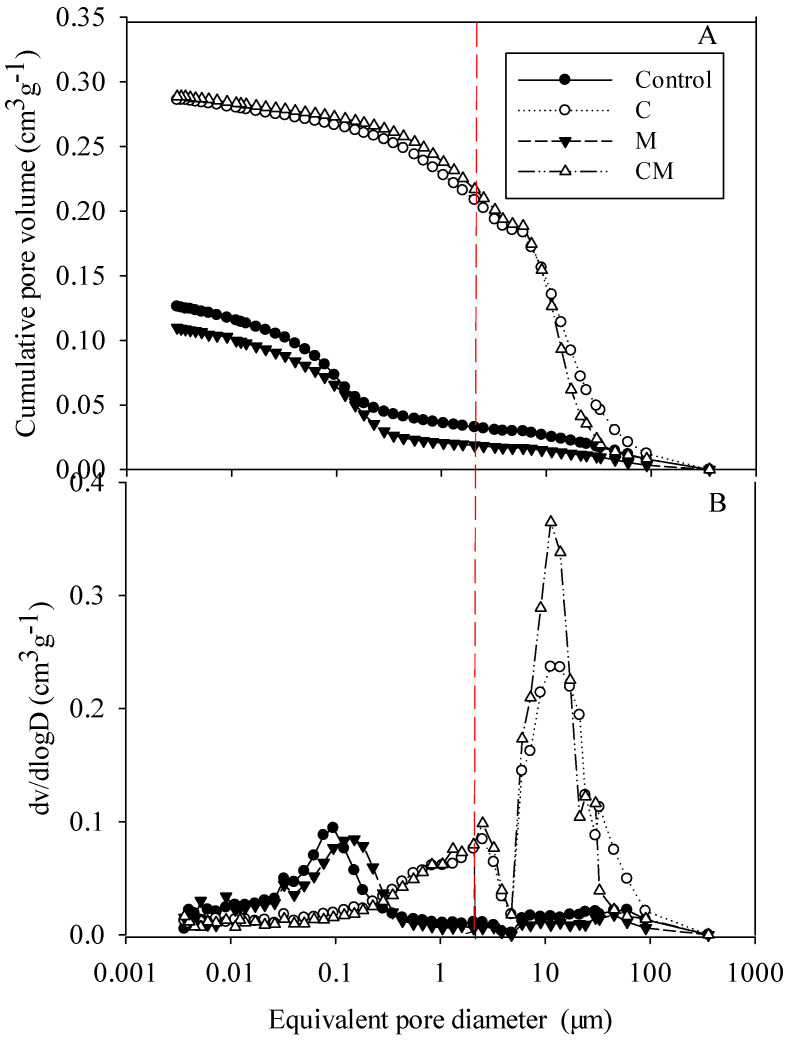
Cumulative (**A**) and differential (**B**) pore size distribution of different treatments determined by MIP in the range of 0.001–1000 μm. The red vertical line is the dividing line (2 μm) between the ineffective and effective pores.

**Figure 3 ijerph-19-11335-f003:**
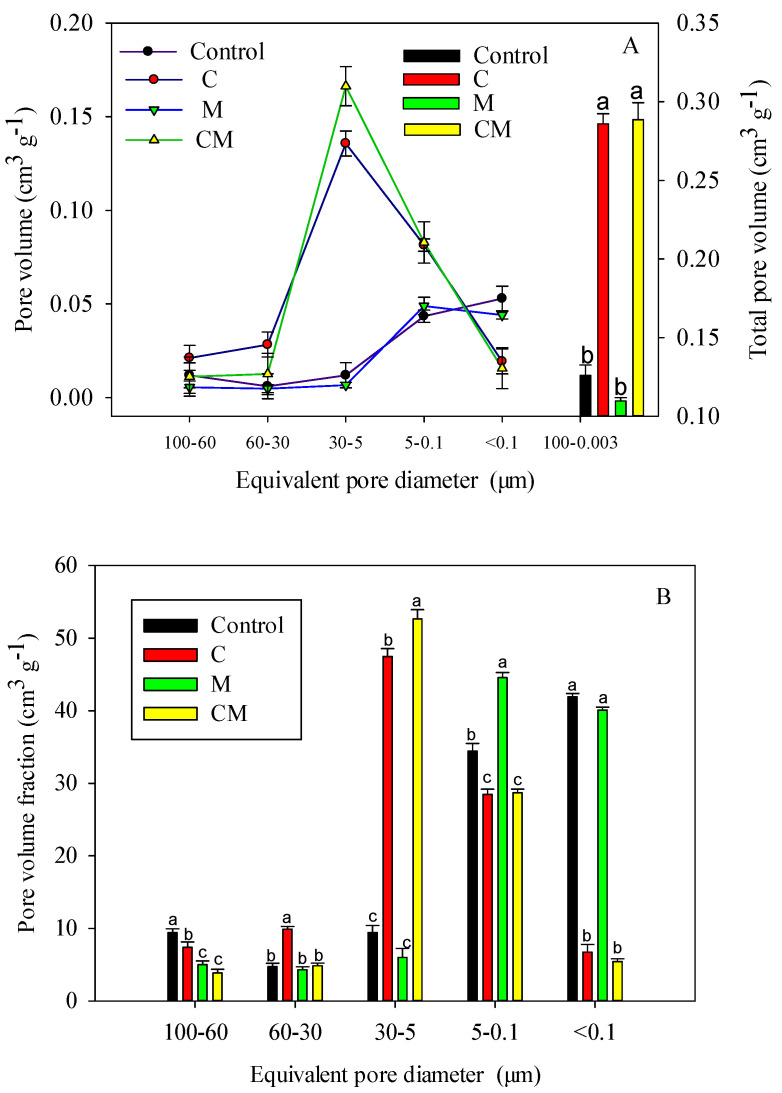
Determination of soil pore volume distribution through mercury intrusion (MIP) using the pore classification method of Cameron and Buchan [33,34]. (**A**,**B**) represent the pore volume and total pore volume at different treatment equivalent pore sizes and pore volume fraction, respectively. Values followed by different letters within horizontal row are significantly different (*p* < 0.05).

**Figure 4 ijerph-19-11335-f004:**
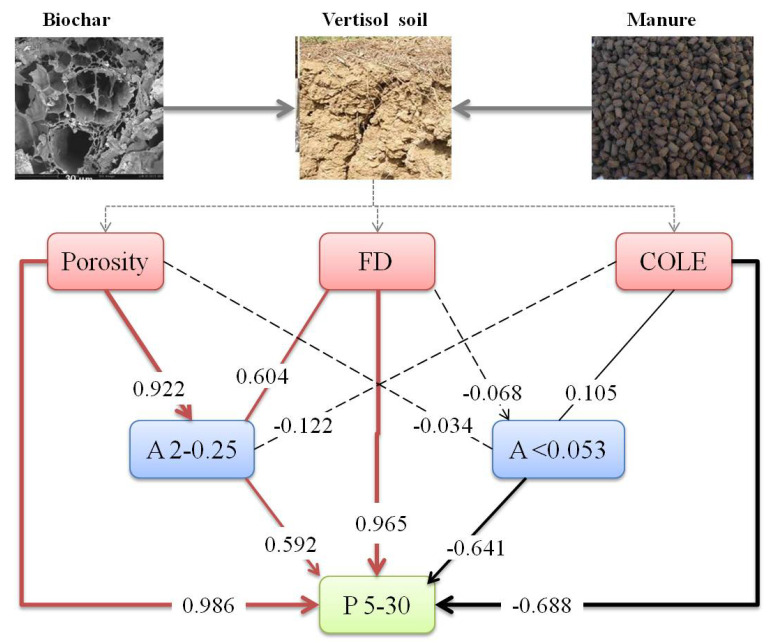
Structural equation modeling (SEM) analysis of the biochar and manure synergistic effects on micropore volume with equivalent pore diameter (5–30 μm).

**Table 1 ijerph-19-11335-t001:** Selected physicochemical properties of vertisol.

Parameter	Vertisol
Sand (2–0.02 mm, %)	26.0
Silt (0.02–0.002 mm, %)	30.7
Clay (<0.002 mm, %)	43.3
Porosity (%)	37.7
Total carbon (g kg^−1^)	5.92
C/N	10.3
CEC (cmol (+) kg^−1^)	23.7
pH	7.50

The pH was determined using the ratio of solid material to water of 1:2.5; particle size distribution was determined using sieving and the pipette method; cation exchangeable capacity was determined using the ammonium saturation and distillation methods; total carbon was estimated through potassium dichromate oxidation and titration with ferrous sulfate.

**Table 2 ijerph-19-11335-t002:** Selected chemical characteristics of the biochar and manure used in the test.

Parameter	Biochar	Manure
Porosity (%)	78.4	-
pH	9.20	8.60
Total carbon (g kg^−1^)	647.2	442.5
CEC (cmol (+) kg^−1^)	41.7	36.2
Phosphorus, mg kg^−1^	2.6	5.3
Potassium, cmol_c_ kg^−1^	4.5	4.9
Calcium, cmol_c_ kg^−1^	3.3	2.0
Nitrogen, %	-	3.1

The pH was determined by the ratio of solid material to water of 1:2.5; particle size distribution was determined by sieving and the pipette method; cation exchangeable capacity was determined using the ammonium saturation and distillation methods; total carbon was estimated by potassium dichromate oxidation and titration with ferrous sulfate. “-” indicates undetermined.

**Table 3 ijerph-19-11335-t003:** Selected physical and chemical properties.

Treatment	pH	SOC (g/kg)	COLE	Porosity (%)	Fractal Dimension
Control	8.18 ± 1.08 a	10.47 ± 0.61 c	0.12 ± 0.74 a	24.14 ± 0.09 b	2.79 ± 0.01 b
C	8.07 ± 0.31 b	13.14 ± 0.26 a	0.10 ± 0.26 c	40.44 ± 0.15 a	2.89 ± 0.01 a
M	8.01 ± 0.14 b	11.87 ± 0.16 b	0.13 ± 0.16 b	21.31 ± 0.08 c	2.80 ± 0.01 b
CM	7.98 ± 1.04 b	12.66 ± 0.63 a	0.09 ± 0.80 c	40.63 ± 0.15 a	2.91 ± 0.01 a

Mean ± SD; n = 4. Values followed by different letters within a column are significantly different (*p* < 0.05).

**Table 4 ijerph-19-11335-t004:** Correlation between micropore volume and equivalent pore diameter (5–30 μm) and different varieties.

Variety	Correlation
A > 2	0.263
A2–0.25	0.601 *
A0.25–0.053	−0.115
A < 0.053	−0.636 *
MWD	0.432
SOC	0.497
pH	−0.866 **
COLE	−0.760 **
P > 60	−0.414
P30–60	0.23
P0.1–5	−0.844 **
P0.01–0.1	−0.986 **
*p* < 0.01	−0.983 **
TPV	0.985 **
Porosity	0.986 **
Fractal dimension	0.965 **

Percentage of A > 2 mm aggregate; percentage of A2–0.25 and 2–0.25 mm aggregates; percentage of A0.25–0.053 and 0.25–0.053 mm aggregates; percentage of A < 0.053 and <0.053 mm aggregates; MWD, mean weight diameter; SOC, soil organic carbon of bulk soil; pH, calculated using the ratio of solid material to water of 1:2.5; COLE, linear expansion coefficient, *p* > 60, macropore volume; P30–60, mesopore volume; P5–30, micropore volume; P0.1–5, ultramicropore volume; P0.01–0.1 and *p* < 0.01, cryptopore volume; TPV, total pore volume. ** Highly significant *p* < 0.01. * Significant at *p* < 0.05.

**Table 5 ijerph-19-11335-t005:** Effects of application of 20 t ha^−1^ biochar and 5 t ha^−1^ Poultry manure on summer maize yield and economic benefits (cost, income, and Profit).

Treatments	Grain Yield		Cost	Income	Profit
	kg (Per Plant)^−1^	kg ha^−1^	CNY		
Control	0.19	10,191.35	5100	15,287	10,187
C	0.37	19,652.48	23,100	29,479	6379
M	0.29	15,277.50	7600	22,916	15,316
CM	0.33	17,435.25	16,600	26,153	9553

## Data Availability

The datasets generated and/or analyzed during the current. Study are available from the corresponding author on reasonable request.

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
