# Peer review of "Effects of Biochar and Manure Co-Application on Aggregate Stability and Pore Size Distribution of Vertisols"

_ijerph, 2022, doi:10.3390/ijerph191811335_

Round 1

Reviewer 1 Report (Previous Reviewer 3)

Answers and responsed to my comments were satisfactory

Author Response

Thanks for your comments.

Reviewer 2 Report (Previous Reviewer 2)

Dear Editor

I am satisfied with all the changes incorporated in the manuscript. The manuscript can be accepted after minor English language corrections. I request you author to kindly send the manuscript to some native English speaker of your field area for the English language correction

Author Response

Thanks for your comments. We have sent the manuscript out for english editing.

This manuscript is a resubmission of an earlier submission. The following is a list of the peer review reports and author responses from that submission.

Round 1

Reviewer 1 Report

The manuscript investigated the synergistic effect of peanut husk derived biochar and poultry manure for the improvement of soil aggregate stability and pore size distribution. While there are lot a studies present related to the sole effect of biochar and poultry manure on soil properties, the combined effect of biochar and manure is rare and the manuscript has the required novelty. The authors did a great job presenting their results in nicely done figures and the manuscript was very well-written and easier to follow. I have only few comments and clarification needed to further improve and I think this manuscript can be considered for publication after addressing these comments.

Abstract: Please extend the terms ‘COLE’ and ‘SOC’ at their first instances (Line no. 29-30).

Introduction: It is written in Line no. 51-52 that the combination of biochar and manure is used to reduce the quantity of biochar as the cost of biochar is a limiting factor. However, the authors reported that the improvement reduces in CM treatment as compared to C treatment. Author should also comment in conclusion section whether to use CM or C in terms of cost and benefit.

Materials and Methods: Line no. 123, Lm and Ld value of wrongly expanded. Lm should be the length of moist soil and Ld should be length of dry soil.

Line no. 125: Incomplete sentence. Please rewrite.

Results: Line no. 165: It seems that authors have wrongly written this sentence. As per Table 2, COLE value in C and CM treatements are lower (not higher) than control soil. Please check.

Figure 2: How the authors have decided whether to use linear equation or quadratic equation to propose relationship between MWD and COLE, SOC?

Figure 6: The proposed linear relationships are not at all suitable for describing the experimental data even for higher R2 value.

There are quite a few errors in citing reference in the text (Line no. 221-222; 270; 305-306; 364-365; 372-374; 413-414). Please correct it.

Also, proof read the revised manuscript one more time before submission. I have noticed few grammatical mistakes in the manuscript.

Reviewer 2 Report

The current study entitled “Biochar and manure effects on aggregate stability and pore size distribution of expansive clayey soil (Vertisol)” is good. For a better understanding in-depth, it is a need time to work on this topic. Furthermore, the achievement of potential benefits by using current technology is also dependent on extensive research work for more exploration. Although the experiment is well organized, I suggest a major revision due to the following deficiencies.

Abstract

  1. Make the title a simple statement. Clay soil is a term but I am confused about what is clayey.
  2. Give the problem statement in a single line.
  3. Give a reason for the selection of the current technique i.e., biochar and manure. There are many other good technologies present then why authors have focused on this technique. Give a statement in a single line.
  4. Quantitative data is also important to support your conclusion. Would you please provide some quantitative data in terms of percentage significant increase or decrease in the abstract?
  5. Please provide a conclusive conclusion with is withdrawn through research in a single line.
  6. Give future prospective in a single line.
  7. As per standard suggestions, please avoid using title words as keywords.

Introduction

  1. Please follow the title and rewrite the introduction in the following sequence i.e., Biochar, manures, aggregates stability, pore size in soil, the importance of vertisols, problem statement, aims of study and hypothesis.
  2. Also, provide a novelty statement at the end. What new things authors have done or correlated in this research compared to old ones?
  3. Would you please give a single line about the knowledge gap which your research has covered along with the hypothesis statement?

Material and methods:

  1. Authors have declared that total soil carbon was 11.5 g/kg and soil is poor in organic carbon. If I am not wrong it is 1.15%. I think for any soil more than 1% organic carbon is considered a sufficient amount when it is utilized for agriculture purposes. Please check it critically.
  2. No protocol details or references are provided for the methods which were followed for the analysis. Please provide that in detail.
  3. Please provide the method of soil sampling and storage which was used in the current study with reference. Method of soil sampling directly affects soil physicochemical attributes that's why it is important to know about that.

Results and Discussion.

    14. What is +/- in table 2. Either it is SE or SD. Please clear that.

    15. Discussion part is fine.

Conclusion

  1. Add the targeted beneficiary audience who will get benefits from this research.
  2. Also, give clear-cut recommendations while describing the best treatment.

Reviewer 3 Report

I run a similarity (plagiarism) and result is to high 53% high percentage from the article: https://doi.org/10.1016/j.catena.2013.10.014 

- Abstract is too long  - Reference in page 2 (Whalen et al 2000) was not mentioned in the reference list and was not given a number - A very long article with many references as a review article.